# Design and Validation of the Scale to Measure Aquatic Competence in Children (SMACC)

**DOI:** 10.3390/ijerph17176188

**Published:** 2020-08-26

**Authors:** Juan Antonio Moreno-Murcia, Luciane de Paula Borges, Elisa Huéscar Hernández

**Affiliations:** Department of Sports Sciences, Sports Research Center, Miguel Hernández University, 03202 Elche, Alicante, Spain; ldepaula@umh.es (L.d.P.B.); ehuescar@umh.es (E.H.H.)

**Keywords:** early childhood, aquatic environment, development, evaluation instrument, measure

## Abstract

(1) Background: The aim of this study was to design and analyze the validity of the SMACC (Scale to Measure Aquatic Competence in Children) to evaluate aquatic competence in three- to six-year-old children. In addition, the relation between real competence obtained with the SMACC and perceived aquatic competence was verified as well as its differences according to sex and age. (2) Methods: Content validation was performed through the consensus of nine experts using the Delphi technique, and comprehension validity was determined through a pilot study on a sample of 122 children. An exploratory and confirmatory factor analysis was performed with two independent samples of 384 and 444 school children between three and six years old, respectively. (3) Results: After the pertinent adjustments, the final questionnaire comprised 17 items, which showed a good fit for both comprehension and content validity. The results of the exploratory and confirmatory analyses support the use of three dimensions in aquatic competence: motor, socio-affective, and cognitive. The correlations support construct validity showing a positive relation with perceived aquatic competence. (4) Conclusions: These promising validity data are discussed from a global and integrative perspective in relation to the improvement of children’s development in the aquatic environment during the early stages of their lives.

## 1. Introduction

Currently, the early childhood stage is receiving an increasing amount of attention due to a consensus among the scientific community that children are capable of learning from birth. Thus, beyond the maturation parameters, there is a special interest in understanding the variables associated with their interaction with the context they evolve in as a means to optimize their development from an integral perspective.

In this sense, the aquatic environment is ideal for putting into practice children’s first psychomotor skills. It contributes to their acquisition of the notion of body through the perceptive components involved in corporal recreational experiences, whereby a child progressively acquires an increasing level of competence in different developmental environments. Specifically, aquatic competence is defined as an integrative concept, where the development of motor patterns and motor skills coexist with other cognitive and socio-affective skills which are necessary to full development [1].

On the other hand, although approximate age groups exist, which helps in the pedagogical adjustment of aquatic educational programs for each evolutionary stage, more and more scientific literature recommends the idea of focusing on the individual differences in children in order to adjust the pedagogical objectives of aquatic activity programs to the specific needs of each child. In this sense, during this stage, measurement is a key aspect as it can verify the advances achieved by a child, and thereby help them move towards a full development in this medium. However, although instruments dedicated to measuring motor and perceptual-motor patterns are well-known in the dry-land [2] and have been supported by a wide theoretical and empirical basis since the 1970, we find that, despite recognition of the idiosyncrasy of aquatic motor competence in children years ago by numerous authors [3,4], today there are very few validated instruments for assessing early childhood in the aquatic environment.

### 1.1. Dimensions of Aquatic Competence

For decades, specialists in the area of aquatic activities have felt the need to change the hegemonic expression “swimming” used for every aquatic activity [3,5,6] with respect to the prevention of drowning. Not all of them have considered this notion in the same way and with the same intentions. For some, this focus should be directed at educating some citizens aquatically, so they can enjoy the water without drowning [7,8]. Meanwhile, for others [1], aquatic competence would refer to the set of knowledge, procedures, attitudes, and affects that people need to solve problems or enjoy different aquatic environments. This definition coincides with those highlighted by specialists in the aquatic world and for those who believe aquatic competence involves knowledge, skills, and values [7]. Thus, Morán [9] defined it as the sum of all aquatic movements that help to prevent drowning and which are associated with an individual’s knowledge, skills, and values, a definition which Stallman [7] and Stallman et al. [8] also adhered to.

The development of perceived competence is a phenomenon linked to the evolutionary development experienced during the different stages of maturation periods [10], which develop through the interaction of perceptive-cognitive, motor, psychological, and social dimensions throughout people’s lives [11]. For this reason, it is necessary to differentiate between two components of aquatic competence. First, perceived aquatic competence stands out as the psychological dimension that involves the self-concept and beliefs that a young person has about their ability or possibilities of solving a problem, and these are developed throughout their school years [12]. On the other hand, there is real aquatic competence, understood as the ability to adapt and evolve by having a command of both fundamental and complex skills in the aquatic environment, which allow them to solve the problems presented or that may arise spontaneously in the water, together with the ability to engage [13], beyond simply preventing and/or avoiding drowning [8]. There are certain moments in development, such as in the ages between three and five, when there are disconnections between real competence and perceived competence because, through trial and error and exploration, they discover possible limitations. In this sense, there are few studies that focus on this aspect. At the age of three, children will use both hands indistinctly, without demonstrating a dominance until they reach four years when it will start to become more manifest and become reinforced over the years. Later, the basic aquatic skills phase will begin, where they will gradually develop aquatic motor competence by participating in activities and games proposed by adults [1].

However, the scientific literature indicates that, the greater is the perception of aquatic competence by the participant, the greater is their real aquatic competence [14,15].

### 1.2. Measure of Aquatic Competence

One of the major limitations that exist to correctly measure aquatic competence in its two dimensions is the lack of validated instruments that are adapted to the different abilities corresponding to the different phases of the early evolutionary stages. Measures of aquatic competence as perceived by children include the Pictorial Scale for Evaluation of Aquatic Competence [12] and the Pictorial Scale of Perceived Water Competence [16]. However, although different authors have researched motor evolution in the aquatic environment and its development [17,18,19] and have created some tools for this, in all cases, they respond to quantitative measures which only focus on one aspect of development, almost always centered on human beings’ affinity with the aquatic environment. Some of these instruments are the Aquaticity Test [20], the Scale of Aquatic Motor Competences [21], the Inventory of Evolutionary Aquatic Development [22], or the Swimming Competence Questionnaire for Children [23]. Although these scales focus on the acquisition of aquatic motor skills, few of them address aquatic competence holistically by considering all the dimensions involved in the effective development of children during their competence experiences in the aquatic environment (cognitive, motor, social, and emotional). Some proposals [1] do contemplate this view, but, because they focus on a maximum period of a year, they cannot be used continuously over several years to be able to verify competence longitudinally.

### 1.3. This Study

The aim of the study was to design and validate an instrument that reliably and validly evaluates aquatic competence from three to six years. More specifically, the objectives of this study were: (a) Design, analyze, and validate the SMACC (Scale to Measure Aquatic Competence in Children) to evaluate aquatic competence in boys and girls aged three to six. To do this, the following steps were established (Studies 1 and 2): determining content validity by means of expert opinion using the Delphi method [24]; confirming comprehension validity of the scale in a pilot study on an independent sample; examining the reliability of the questionnaire; and corroborating the construct’s tri-dimensionality. (b) Confirm its relation to perceived aquatic competence (Study 3), and examine the relation of real aquatic competence according to age and sex.

On the basis of the above arguments, the expectations were to obtain an instrument consisting of three dimensions that can measure aquatic competence in 3–6-year-old children, which shows that, the greater is the perception of aquatic competence, the greater is the real aquatic competence and presents differences for age but not for sex.

### 1.4. Study 1

The aim of this first phase was to create an instrument to measure aquatic competence and to obtain content and comprehension validity.

## 2. Materials and Methods

### 2.1. Ethics Statement

This study has been approved by the Research Ethics Committee of Universidad Miguel Hernández de Elche (Elche, Spain) (2019.286.E.OEP) and meets all ethical and legal standards that are applicable to the research of this survey modality.

### 2.2. Participants

The Delphi method was used as a strategy to evaluate the instrument of aquatic competence. Two groups of humans were formed to validate the designed instrument, establishing, in this case, a coordinating group and an expert group. The coordinating group had a good knowledge of the Delphi method and comprised three academic researchers who were familiar with the subject and were able to intercommunicate easily [25]. The group of selected experts, who have close contact with this problem and wide experience, comprised nine university lecturers and researchers of renowned prestige in the area of aquatic competence [26].

### 2.3. Procedure

The methodological sequence was established in three phases: initial, exploratory, and final.

Initial phase. The coordinating group was responsible for: defining the research problem; selecting the group of experts and obtaining their commitment to collaborate; interpreting the partial and final research results; and supervising its correct progress, being able to make adjustments and corrections.

Exploratory phase. In this phase, the questionnaire was designed in its experimental version for the final version to be determined (Table A1). To do this, the first version underwent a first round of analysis and discussion by the members of the coordinating group, and certain corrections and adjustments were made according to the qualitative criteria that had obtained a majority consensus. The agreed version was validated in a second round by the expert group to gather information on the most stable quantitative and qualitative criteria. The steps are outlined as follows: (1) selection of experts whose contribution to the study is considered invaluable; (2) invitation to participate in the process by email; (3) sending and receiving the questionnaire by electronic mail in an attached file, consisting of a first page with a brief introduction and explanation of the research subject, a page for the respondent to fill out with their personal details, a clear description of the study objective, and the instructions for completing the questionnaire, followed by the corresponding instrument for validation; (4) validation instrument, namely a Likert type scale with four categories according to adequacy, clarity, coherence, and relevance or pertinence to the dimension to be researched, as well as an open question to obtain qualitative valuations about the items or the introduction of any new ones, with a maximum deadline of 30 days; (5) follow-up by email of the selected people; (6) collection of the completed scales; and (7) analysis of information contained in the Delphi scale. The results of this consultation were analyzed by the coordinating group from a quantitative and qualitative point of view, drawing on the opinions expressed by the experts in response to the open questions included in the consultation instrument.

Final phase. In the last phase, the results from the whole validation process of the final version were synthesized to be subsequently applied in Study 2.

This study complied with the ethical standards and values required for research with humans (informed consent, rights to information, personal data protection and guarantees of confidentiality, non-discrimination, and the possibility of abandoning the study at any stage). A favorable report was received from the Ethics Committee of the University Miguel Hernández of Elche (project nº 2019.286.E.OEP).

### 2.4. Data Analysis

Qualitative data were analyzed through content analysis. For quantitative data, the preparatory data analysis and calculation of descriptive statistics were performed with the software program SPSS 25.0.

## 3. Results

### 3.1. Construction of the Instrument

Once the limitations of the instruments available were analyzed, the SMACC was drawn up. Content was determined through bibliographical review and with reference to expert opinion [27]. An initial bank of items was created from different questionnaires and scales for evaluating aquatic competence as well as other dimensions of personal psychological and social development social. A first experimental version was constructed dividing responses to each question into five levels of difficulty, which were specific to each item.

The coordinating group went on to read and classify each item according to the dimensions: motor, socio-affective and cognitive. Choice was made according to adequacy, rational criterion, and ages of target sample, resulting in a bank of 25 items. The items included in this version were taken verbatim from the original questionnaires, reformulated, or written specifically for the occasion.

To reach optimum levels of content validity, the expert technique was used, and a pilot study was carried out to verify content validity from the perspective of the comprehension validity by the subjects of the study. The experts were asked to value different aspects of the initial information, the questionnaire, the items, and the global valuation of each one [28], taking into account the degree of comprehension, the appropriateness of the wording, etc.

With respect to the items, the degree of pertinence to the subject of the study and to what extent each of them should be included was recorded on a scale of 1–4. It was decided that all the items with mean values close to 2 should be eliminated, items with values around 3 should be modified, and values close to 4 should be accepted. Once these calculations were made, 23 items were decided on for selection.

To verify the comprehension validity of the instrument, a pilot study was carried out. After the questionnaire had been administered by seven experts to a total sample of 122 boys and girls (lasting between 8 and 10 min), the degree of comprehension was analyzed from a qualitative point of view, recording questions, doubts, and suggestions made by the participants (teachers and students) during the session.

The qualitative contribution was completed with the quantitative contribution of the mean scores given by the experts for each item. The results were analyzed including the valuations by the coordinating group and the expert group, constituting two independent sources, which guaranteed the adequacy of the instrument. Out of the 23 items that were initially included in the questionnaire, 13 did not undergo any modification, since they obtained values close to 4 and the experts did not suggest another version. Three items with values around 2 were eliminated and substituted by new ones following the recommendations from the expert group. The remaining seven items, with values around 3, were modified in accordance with the experts’ opinion, and their final formulation was agreed on by the coordinating group. The exploratory structure was based on the experts’ assessment with respect to content validity. Therefore, once the items were classified according to their corresponding factors, the next phase was carried out to verify the exploratory factor structure and analyze reliability.

### 3.2. Study 2

The aim of this second phase was to analyze the exploratory factor structure of the scale.

## 4. Materials and Methods

### 4.1. Participants

The sample comprised 384 children, 195 boys and 189 girls Their ages ranged between 3 and 5 years old, with a mean age of 4.02 (SD = 0.82). Distribution by age was as follows (3, *n* = 126; 4, *n* = 123, 5, *n* = 135). With respect to aquatic experience, the 3-year-olds had none, the 4-year-olds had spent a year doing aquatic activities (one day a week, throughout the school year), and the 5-year-olds had had two years of experience (one day a week during two school years).

### 4.2. Measures

#### Aquatic Competence

We used the SMACC from Study 1. It contains 23 items grouped into three dimensions; socio-affective, consisting of seven items; cognitive with seven items; and motor, consisting of nine items. The children’s behaviors were evaluated using a five-point rubric. For example, for the item that corresponds to breathing (“when the children in the shallow end were asked to make bubbles under the water by releasing air through the mouth and nose…”): 1 corresponds to “Blows without touching the water with their face”; 2 to “Blows by only putting their mouth at the level of the water”; 3 to “Doesn’t blow in the water, but puts their face completely in the water”; 4 to “Blows through the mouth and nose, putting their face completely in the water”; and 5 to “Is able to coordinate breathing (takes in air and releases it continuously and several times)”.

### 4.3. Procedure

The sports installations responsible persons who accepted to participate, along with the aquatic instructors, were informed about the research objectives and the activities to be evaluated. One researcher from the coordinating group personally evaluated each of the children, going through the different items from the questionnaire while observing the classes, without influencing class dynamics or development. Participation was voluntary and anonymity was preserved by allocating each child with a numerical code and geographical area. Parents were previously informed about the nature of the study and signed a consent form. A favorable report was received from the Committee for Responsible Research (OIR, project *No.* 2019.286.E.OEP). Observation time for each child was approximately 15 min.

### 4.4. Data Analysis

An exploratory factor analysis (EFA) was performed to establish the instrument’s factor structure, and the internal consistency of the instrument was analyzed using Cronbach’s alpha. Alpha values greater than or equal to 0.70 were considered good. There are alternative ways to estimate the internal consistency of an instrument that were not considered in this study, such as the halving method or the Kuder–Richardson method. Cronbach’s alpha was chosen because it is included in the software used. Data analysis was carried out with the statistical software SPSS 25.0.

## 5. Results

An exploratory factor analysis of the main components was performed with oblimin rotation. After a first analysis, seven of the items did not reach the established minimum saturation (0.40), and they were eliminated. Another analysis was made, where the 17 items were grouped into three areas (Table 1): socio-affective with five items, cognitive with five items, and motor with seven items. It was decided to force the grouping into three dimensions because these were the dimensions the theoretical review of studies indicated the vision of aquatic competence should contain. These three factors obtained eigenvalues greater than 1 (7.44, 4.27, and 2.49, respectively), explaining a total variance of 83.60% (43.81%, 25.14%, and 14.65%, respectively).

### 5.1. Analysis of Internal Consistency

Cronbach’s alpha coefficient obtained for each of the dimensions was 0.95 for the motor dimension, 0.95 for the cognitive dimension, and 0.93 for the socio-affective dimension.

### 5.2. Study 3

The objective of this third phase was to carry out a confirmatory factor analysis with the SMACC and to show its relation to perceived aquatic competence.

## 6. Materials and Methods

### 6.1. Participants

The sample comprised 444 children, 235 boys and 208 girls. Their ages ranged between 3 and 5, with a mean age of 4.45 (*SD* = 0.84). Distribution by age was as follows: 3, *n* = 134; 4, *n* = 149; and 5, *n* = 161. With respect to aquatic experience, the 3-year-olds had had experience from that school year, the 4-year-olds had spent two years doing aquatic activities (one day a week throughout the school year), and the 5-year-olds had had three years of experience (one day a week during two school years).

### 6.2. Measures

#### 6.2.1. Aquatic Competence

We used the final version of the SMACC from study 2. It consisted of 17 items grouped into three dimensions (Table A1): socio-affective with five items, cognitive with five items, and motor with seven items. Cronbach’s alpha coefficient was 0.92 for the motor dimension, 0.93 for the cognitive dimension, and 0.91 for the socio-affective dimension.

#### 6.2.2. Perceived Aquatic Motor Ability

We used the factor “aquatic motor ability” from the Pictorial Scale of Perceived Aquatic Competence (PSPAP) by Moreno and Ruiz [12], which measures the level of aquatic motor ability that a child perceives (Figure 1). The six items are answered on a Likert scale of three options (represented by three comic images), where A corresponds to “better”, B to “moderate”, and C to “worse”. Each alternative was presented individually to the child with three comic images to facilitate their understanding of the question. The child had to point with a pencil to which of the images seemed more like themselves. To control possible sources of error, the intra-element (response option) was presented in random order, which was varied per item. Internal consistency for this dimension was 0.85.

### 6.3. Procedure

To gather information, the same procedure as described in Study 2 was followed.

### 6.4. Data Analysis

A confirmatory factor analysis (CFA) was carried out to confirm the structure of the instrument). To check the validity of the measurement model, the following coefficients or goodness of fit indices were considered: χ^2^, χ^2^/gL, RMSEA (Root Mean Square Error of Approximation), and SRMR (Standardized Root Mean Square Residual), as well as incremental indices CFI (Comparative Fit Index) and IFI (Normed Fit Index). These goodness of fit indices are considered acceptable when χ^2^/gL is less than 5, incremental indices (CFI and IFI) are greater than 0.90, RMSEA error rate is less than 0.08, and SRMR error rate is less than 0.05. Similarly, the internal consistency of the instrument was analyzed by Cronbach’s alpha and the descriptive statistics were obtained (mean and standard deviations) as well as the bivariate correlations of all the variables. To verify the relation between real and perceived aquatic competence according to age and sex, a MANOVA was performed. To examine the relation of real aquatic competence (contemplating in one measure the mean obtained from the dimensions of motor, cognitive and socio-affective) according to sex and age, a differential analysis was performed with real aquatic competence as dependent variable and age and sex as independent variables. Data analysis was carried out using the statistical software SPSS 25.0 and AMOS 25.0

## 7. Results

### 7.1. CFA

The factor structure was analyzed using a confirmatory factor analysis with the 17 items included in the three-factor model (motor, cognitive, and socio-affective). The maximum verisimilitude estimation method was used together with the bootstrapping procedure. Since the result of the Mardia multivariate coefficient was 286.63, which indicated a lack of multivariate normality of the data, robust maximum verisimilitude estimation was used. Based on the modification indices, ten indications of standard error were established and a new analysis was performed, the results of which showed a better fit of the model (χ^2^ (62, *n* = 444) = 554.96, *p* = 0.000; χ^2^/d.f. = 2.90; CFI = 0.90; IFI = 0.90; RSMR = 0.04; RMSEA = 0.05). The model proposed presented a reasonable approximation to the data and contributed to supporting the hypothesis of the multidimensionality of the construct. The estimations of the factor saturations for each of the items in their respective factors are illustrated in Figure 2.

### 7.2. Descriptive Analysis and Bivariate Correlations

The socio-affective dimension presented the best result out of the three dimensions of the SMACC followed by the cognitive area and the aquatic motor area. Perceived aquatic competence showed a mean of 2.49 out of 3. All dimensions correlated positively with each other (Table 2). Figure 3 shows the relation between real aquatic competence (contemplating in one measure the mean obtained from the dimensions of motor, cognitive and socio-affective) and perceived competence, according to the data from the linear regression.

### 7.3. Differential Analysis of Real Aquatic Competence by Sex and Age

No differences were found for sex (Wilks’ Lambda = 0.99, *F* (2, 379) = 0.42, *p* > 0.05, η2 = 0.00), but differences were found for age (Wilks Lambda = 0.34, *F* (4, 758) = 839.84, *p* < 0.001, η2 = 0.81) in real aquatic competence (*F* (2, 379) = 0.23, *p* < 0.001, η2 = 0.94) for all ages (Figure 4).

## 8. Discussion

Validated instruments that measure aquatic competence are very scarce, besides having low specificity for ages between three and five. On this basis, the main objective of this study was to design, develop, and validate the SMACC. The theoretical design was confirmed thanks to the adequate psychometry obtained, and the study hypothesis was also confirmed.

The SMACC was therefore adjusted to a model of 17 items grouped into three dimensions (motor, cognitive, and socio-affective). The motor area included seven measures that evaluate a child’s ability to use and control behaviors of movement, manipulation, turns, space-time perception, immersion, and breathing. The cognitive area consisted of five situations that explore conceptual skills, valuing perceptive discrimination, memory, and reasoning. The socio-affective area comprised five measures that mainly evaluate a child’s competence in establishing significant social and emotional interactions, their attitude towards the task presented, and their relationship with adults and peers. The three dimensions established in the instrument permit the valuation of different aspects related to competence that are similar to other instruments used for other age groups [20,21,22,23].

From these first promising results of validity of the SMACC, this scale can be seen as a useful instrument for measuring aquatic competence in the different ages that correspond to the later stage of early childhood. In addition, it has also served to verify that there are no differences according to sex [1], which indicates that the instrument would be adequate for measuring boys and girls indistinctly.

The results about the relation between real and perceived competence are in line with the previous research. Moreno-Murcia, Huéscar, and Parra [15], verified the existence of a positive relation between real and perceived aquatic competence in early childhood. In this sense, it is common for children aged between three and six to show a variety of fears that can make it difficult and even impossible for them to develop certain actions that they would in reality be able to perform [14]. For this reason, an aquatic teacher would be responsible for reinforcing a positive self-concept in the children by using strategies that promote the development of perceived aquatic competence, making it possible for them to perform activities that they did not believe they were able to do before, and thereby improve both dimensions of competence.

It is important to point out that this instrument should be used by professionals in this field, such as aquatic teachers and swimming monitors who are responsible for carrying out and supervising its application. Furthermore, this scale is aimed at children aged between three and five inclusive; it is not a suitable instrument for use with other age groups. It can also be used to measure the advances made in aquatic competence in ages between three and five, providing information about their level of aquatic competence. For this reason, this instrument helps to create new study perspectives that focus on dimensions of aquatic competence which to date have not received enough attention for this age group. Likewise, the use of a five-point evaluation system has meant that it was possible to make a sensitive evaluation that considers the skills the participant is beginning to acquire as well as those that have already been wholly acquired, thereby making an individual and personalized evaluation. Future research developments should continue in the line of the replication of the results found in this study with samples from other contexts.

Nevertheless, this study is not exempt from limitations. It is necessary to highlight the importance of performing a greater number of transversal studies in order to replicate the results obtained and to obtain a larger sample size. Longitudinal designs would also be useful for providing more information about the development of aquatic motor competence over time, highlighting the changes that occur. It is also necessary to consider in the next studies the temporal stability of the scale. Finally, it would be of great interest to carry out a transcultural validation of the instrument, and to even be able to determine the possible causes of real competence and children’s perceived competence. In addition, for professionals from the aquatic education context, it would be useful to have a record of the abilities of their learners in order to be able to provide the necessary content at the right moment, where the evaluation of both real and perceived competence would be necessary [29].

## 9. Conclusions

The SMACC was designed, analyzed, and validated to evaluate the aquatic competence in children from three to six years old. A relationship between actual aquatic competence and perceived aquatic competence was confirmed. There are no differences by sex, but by age, in actual and perceived aquatic competence.

## Figures and Tables

**Figure 1 ijerph-17-06188-f001:**
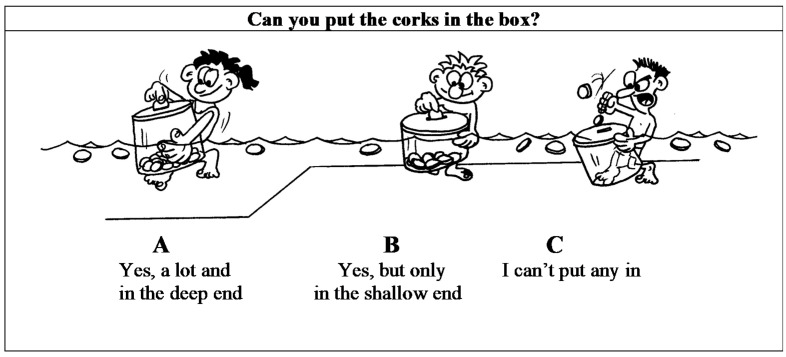
Example of an item from the Pictorial Scale of Perceived Aquatic Competence (PSPAP) by Moreno and Ruiz [12].

**Figure 2 ijerph-17-06188-f002:**
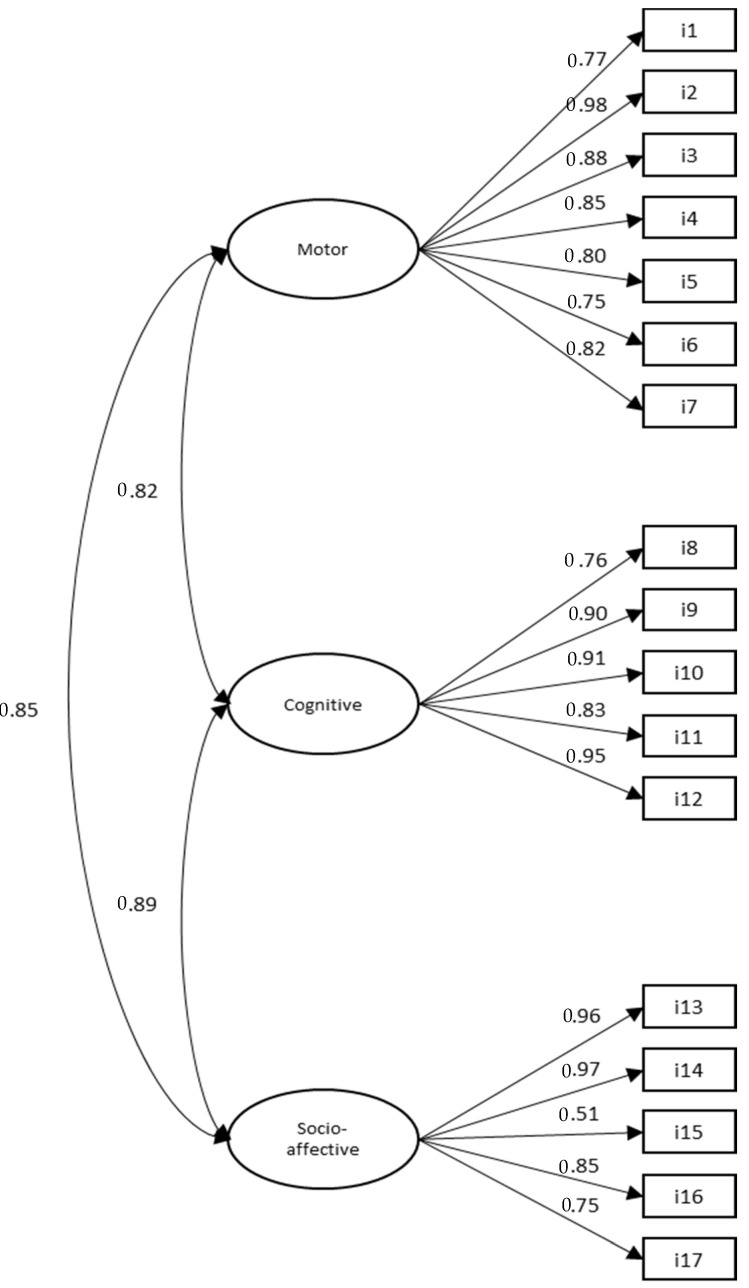
CFA SMACC.

**Figure 3 ijerph-17-06188-f003:**
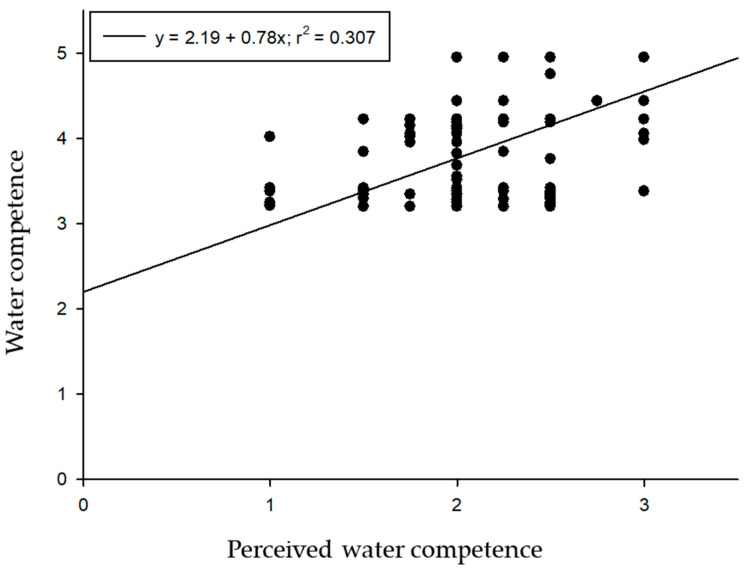
Representation of the relation between real and perceived aquatic competence.

**Figure 4 ijerph-17-06188-f004:**
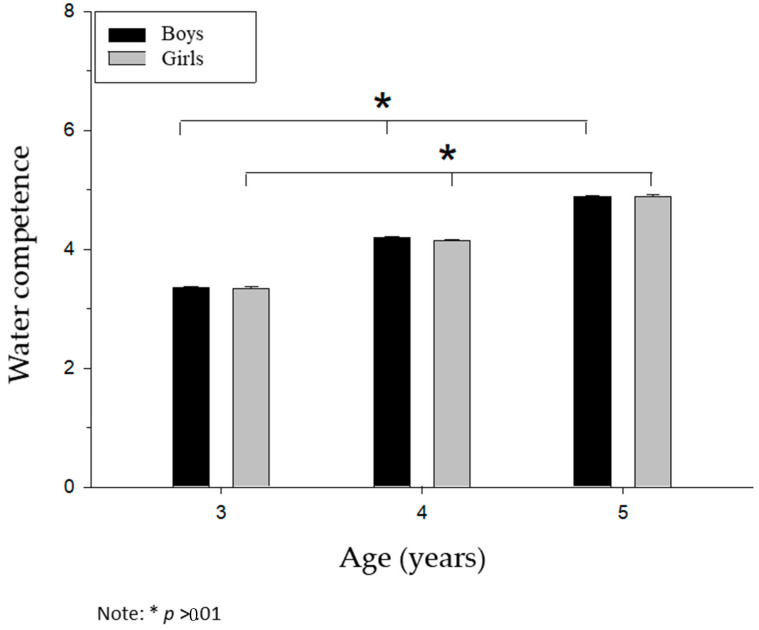
Differential analysis of aquatic competence according to sex and age.

**Table 1 ijerph-17-06188-t001:** Exploratory factor analysis SMACC.

	Motor	Cognitive	Socio-Affective
1. Breathing	0.869	-	-
2. Dorsal Balance	0.819	-	-
3. Manipulation	0.952	-	-
4. Ventral movement	0.911	-	-
5. Turns	0.755	-	-
6. Dorsal movement	0.690	-	-
7. Immersion	0.767	-	-
8. Corporal schema	-	0.882	-
9. Temporality	-	0.715	-
10. Autonomy	-	0.653	-
11. Reasoning	-	0.822	-
12. Oral language	-	0.691	-
13. Communication	-	-	0.534
14. Solving conflicts	-	-	0.921
15. Self-control	-	-	0.593
16. Self-control	-	-	0.478
17. Self-control	-	-	0.397

**Table 2 ijerph-17-06188-t002:** Descriptive statistics and Correlations of all the variables.

Variables	*M*	*SD*	1	2	3	4
1. Motor	3.75	0.99	-	0.98 **	0.91 **	0.57 **
2. Cognitive	3.98	0.75	-	-	0.87 **	0.49 **
3. Socio-affective	4.72	0.23	-	-	-	0.58 **
4. Perceived aquatic competence	2.49	0.46	-	-	-	-

Note: ** *p* < 0.001.

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
