# Peer review of "Design and Validation of the Scale to Measure Aquatic Competence in Children (SMACC)"

_ijerph, 2020, doi:10.3390/ijerph17176188_

Round 1
Reviewer 1 Report
GENERAL COMENT
Very useful and pertinent study. My congratulations to the authors.
SPECIFIC COMENTS
Line 12.
Write out the name for the acronym SMACC
Line 16.
The sentence “… pilot study on two independent samples of 384 and 444 school” is not clear
Line 210.
There is a typing error in “…the 3-year-polds had none…”
Line 215.
It is the first time the scale’s designation appears in full. I suggest that this full designation come in the abstract.
Line 237.
Although internal consistency was determined through Cronbach’s alpha, it is important to report that is not reported an alternative to this procedure considering the five-point evaluation system (similar to Likert-type scale). Therefore, and due to the discussion about the usefulness of Cronbach’s alpha, authors will be protected.
Line 242.
The following sentence should finish with a period instead of a comma: “…they were eliminated,”
“where the 17 items were grouped into three areas”. The authors should explain how this was made.
Line 262.
No need to type the full designation, just the acronym.
Line 267.
6.2.1. It is mentioned the results of internal consistency, but nothing is referred to temporal consistency. Is this scale stable in time?
Line 295.
The quality of adjustment of the model is defined according to quality adjustment indexes and respective reference values in which c2 /df <2-3. Considering that the author mention c2 /df = 8.14, the model is not adjusted. It is recommendable that authors revise and confirm the values obtain in this adjustment indice. Also, it is recommended that authors indicate the Root Mean Square Error of Approximation (RMSEA) considering is one of the most informative indices of Equation Structural Modeling and it is sensitive to the number of parameters estimated in the model
Line 326 and 329
No need to type the full designation, just the acronym.
Author Response
Thanks for the review

Reviewer 2 Report
This manuscript presents a relevant topic to publish in Int. J. Environ. Res. Public Health, which could be accepted with some minor revisions.
In my opinion, the introduction provides adequate information and structure to set up the research questions raised in manuscript; the methodology provides sufficient detail but that can still be an improvement; results and discussion section is sufficiently clear and precise; findings and literature support discussion and conclusion.
After carefully reading your manuscript, I point out some aspects that must be improved and corrected:
- Some aspects of formatting should be corrected (spelling and punctuation, references). Please, correct what it is pointed out in the body of the manuscript;
- The first time that SMACC abbreviation seems in your manuscript is in the abstract (line 12) and the introduction (line 101), but the reader only knows what that means on line 215. Abbreviations must be defined in parentheses the first time they appear in the abstract, main text and figure or table caption and used consistently after that. Please, correct this aspect.
- In the section on methods- data analysis, the authors should introduce cutoff values for a better interpretation of the data. For example, how can we interpret Cronbach's alpha values (α> 0.9 - Excellent, α> 0.8 - Good, α> 0.7 - Acceptable, α> 0.6 - Questionable, α> 0.5 - Weak and α <0.5 - Unacceptable ”. Also, in confirmatory factor analysis (study 3), what were the reference values to consider that the adjustment of the model is acceptable? The authors should try to be more descriptive in statistical procedures.
- Statistical symbols must be in italics (N, n, p, r, F ....).
- (Brazelton, 1973) : line 46 - This reference is not numbered in the manuscript, nor is it included in the bibliography.
- I suggest changing the term "motricity" to "motor" in all manuscript.
- in figure 3, the level of significance of the differences (*), must have a caption.

Author Response
Thanks for the review

Reviewer 3 Report
Dear Authors
Congratulations on your manuscript. I read it with pleasure and found merit in the overall research you have come up with. This tool you presented can be useful to standardize the concept of Aquatic Competence (AC) in different parts of the world; to compare AC; to evaluate the learning-teaching process regarding the main outcome – AC; along with other possibilities you have already mentioned in your final remarks in the discussion section.
Nevertheless, I have some questions that might enhance the comprehension of the readers, and I would like you to consider them. Sorry for the long list, but the manuscript is long itself. In the end, there are some minor questions about English.
- Consider altering the title to have the SMACC defined. Something like this - Design and validation of the Scale to Measure Aquatic Competence in Children – SMACC
- In the introduction:
- The first paragraph is too long and should have one or more references supporting the statements. For instance, the authors state “consensus among the scientific community” and then finishes the paragraph with a self-citation. The same in the second paragraph (“more and more scientific literature recommends…”)
- I prefer to read “aquatic environment” (lines 23, 24, 31, 43, 48, 69, 87, 90, 95, and 380) rather than “aquatic medium”. In line 31 I suggest one of this two possibilities: i) “In this sense, the aquatic context is an ideal environment”; or ii) “In this sense, the aquatic environment is ideal for putting into practice”. Please consider changing.
- In line 44, authors can consider dry-land instead of land medium. If some confusion comes up with dry-land, as it is likely to refer to training, use “environment” as well. It is a suggestion, in any case.
- In line 46, authors refer to (Brazelton, 1973) which is not in the IJERPH references norm neither in the references section. Please update it.
- In line 58 authors refer to Kevin Moran in the definition of AC, but the marked reference [1] refer to Moreno-Murcia. Please correct.
- In 1 Dimensions of AC, authors pointed out several dimensions (perceptive-cognitive, motor, psychological and social dimensions) and then “differentiate between two components of aquatic competence…”; Further, in 1.2 Measure of AC, authors refer to these two components as dimensions (line 82) “to correctly measure aquatic competence in its two dimensions is the lack of validated instruments”. Consider maintaining text linearity.
- In line 92 I think that reference [24] is the [23]. The Swimming Competence Questionnaire for Children must be Chan et al. (2020) and not Dalkey and Helmer (1963).
- The first time authors refer to the SMACC is in line 101 (and the acronym is SCAMM. Please correct) to point out the objectives. Since this manuscript is contributing for the first time to SMACC, there was no need to refer to it in the introduction. However, it should be defined here as it is in line 215 (“We used the Scale to Measure Aquatic Competence in Children (SMACC) from study 1”). Then in line 215 authors can use only SMACC.
- In 2. material and methods:
- Check with the editors if it is possible to have the Ethics statement before the studies description, as it is the same protocol (lines 118 and 233).
- In 2 participants, please check if the references [26] (line 125) and [27] (line 127) are the correct ones. Seems [25] and [26]
- The las paragraph od 3, lines 155-159 is already in 2.1.
- In 3. Results:
- In line 167, check if reference [28] isn’t [27] Crocker and Algina (1986).
- In 4. Materials and Methods:
- Authors can see that in 1.1 AC, is the first time SMACC is referred to. I’ve made my observation before.
- In 5. Results:
- In line 243, the authors refer to “socio-emotional”. In all the text as well as in table 1, figure 1 and table 2 authors refer as socio-affective.
- In 6. Materials and Methods:
- In 6.2.1. AC, there is no need to write Scale to Measure Aquatic Competence in Children. SMACC would be sufficient.
- In line 265 authors refer to Cronbach’s alpha coefficient values. The values presented are not the same as presented in 1 Analysis of internal consistency. Am I missing something, or was it some copy-past error? Please check.
- In 2.2. Perceived aquatic motor ability, since the author is the same, in my opinion, it would add some value if the author could choose the best comic to illustrate that scale and insert the image. Check with editors if it is possible
- In 7. Results:
- In line 288, authors can use only the acronym of CFA. It was already defined in line 280.
- In line 303, use socio-affective instead of socio-emotional.
- In line 307, use mean instead of average as you did in line 314.
- In 3. Differential analysis of real aquatic competence by sex and age, authors besides reporting the results, described the statistical analysis “To examine the relation of real aquatic competence (contemplating in one measure the mean obtained from the dimensions of motricity, cognitive and socio-affective) according to sex and age, a differential analysis was performed with real aquatic competence as dependent variable and age and sex as independent variables.”. Shouldn’t this description be in the 6.4. Data Analysis, line 279?
- In Figure 3, authors should change the legend to boys and girls instead of men and women, as in line 342.
- In figure 3, there are some * in the graph. Please add a legend in the figure or when reporting the differences in the text.
- In 8. discussion:
- In line 329, SMACC would be enough.
- In line 329 seems to me that a verb is missing “The Scale to Measure Aquatic Competence in Children was, therefore, adjusted to a model of 17 items grouped into three dimensions (motor, cognitive, and socio-affective)”. Please chek
- In line 352, if changed before, change again now from dimensions of competence to components of competence. Check my point regarding line 82.
- Line 374, please check the reference. Seems to be [29] instead of [28].
- In 9. Conclusion:
- I suggest authors to redo the conclusion according to the objectives. This paragraph of the conclusion can come in the last part of the discussion. The objectives stated in the manuscript were: a) to design, analyze and validate the SMACC to evaluate aquatic competence in boys and girls aged three to six; and b) to confirm its relation to perceived aquatic competence, and to also examine the relation of real aquatic competence according to age and sex. Please respond only to the objectives.
Congratulations, it is a remarkable and useful instrument. Thank you for the opportunity.
English corrections
Line 30 – “associated with their interaction”
Line 47 – a verb seems to be missing “today there are very few validated instruments for assessing, evaluate, measuring early childhood in the aquatic medium”, or did I miss the point here?
Line 51 – “used for every aquatic activity with respect to”
Line 74-76 – “At the age of three, children will use both hands indistinctly, without demonstrating a dominance until they reach four years when it will start to become more manifest and become reinforced over the years.”
Line 185 – “and values close to 4 should be accepted”
Line 210 – “the 3-year-olds had none”
Line 226 – Check if it sounds better and less wordy “The sports installations responsible persons who accepted to participate, along with the aquatic instructors, were informed about the research objectives and the activities to be evaluated.”
Line 226 – if maintained, “were contacted to be informed about the objective”
Line 324 – consider “Validated instruments that measure aquatic competence are very scarce, besides having low specificity for ages between 3 and 5.”
Line 324 – if maintained, “Validated instruments that measure aquatic competence are very scarce, and in addition, they have low specificity for ages between 3 and 5.”
Author Response
Thanks for the review
